# Novel Spiro-Core Dopant-Free Hole Transporting Material for Planar Inverted Perovskite Solar Cells

**DOI:** 10.3390/nano13142042

**Published:** 2023-07-10

**Authors:** Raquel Royo, José G. Sánchez, Wenhui Li, Eugenia Martinez-Ferrero, Emilio Palomares, Raquel Andreu, Santiago Franco

**Affiliations:** 1Instituto de Nanociencia y Materiales de Aragón (INMA), Departamento de Química Orgánica, CSIC-Universidad de Zaragoza, 50009 Zaragoza, Spain; raquelroyo@posta.unizar.es (R.R.); randreu@unizar.es (R.A.); 2Institute of Chemical Research of Catalonia, The Barcelona Institute of Science and Technology (ICIQ-BIST), Avinguda Països Catalans 16, 43007 Tarragona, Spain; jgsanchez@iciq.es (J.G.S.); wli@iciq.es (W.L.); emartinez@iciq.es (E.M.-F.); 3Catalan Institution for Research and Advanced Studies (ICREA), Passeig Lluïs Companys, 23, 08010 Barcelona, Spain

**Keywords:** spiro-core, inverted perovskite solar cells, hole-transporting material, dopant free

## Abstract

Hole-transporting materials (HTMs) have demonstrated their crucial role in promoting charge extraction, interface recombination, and device stability in perovskite solar cells (PSCs). Herein, we present the synthesis of a novel dopant-free spiro-type fluorine core-based HTM with four ethoxytriisopropylsilane groups (**Syl-SC**) for inverted planar perovskite solar cells (iPSCs). The thickness of the **Syl-SC** influences the performance of iPSCs. The best-performing iPSC is achieved with a 0.8 mg/mL **Syl-SC** solution (ca. 15 nm thick) and exhibits a power conversion efficiency (PCE) of 15.77%, with *J_sc_* = 20.00 mA/cm^2^, *V_oc_* = 1.006 V, and FF = 80.10%. As compared to devices based on PEDOT:PSS, the iPSCs based on **Syl-SC** exhibit a higher *V_oc_*, leading to a higher PCE. Additionally, it has been found that **Syl-SC** can more effectively suppress charge interfacial recombination in comparison to PEDOT:PSS, which results in an improvement in fill factor. Therefore, **Syl-SC**, a facilely processed and efficient hole-transporting material, presents a promising cost-effective alternative for inverted perovskite solar cells.

## 1. Introduction

The need to replace fossil fuels with alternative energy sources has driven increased scientific interest in using solar energy. In contrast to first-generation silicon solar cells, new photovoltaic technologies have been developed to simplify manufacturing processes and reduce infrastructure costs. As emerging technologies, organic and perovskite solar cells show promising potential, and both types of devices are built in similar architectures. Ternary organic solar cells (TOSCs) are being studied in the field of organic photovoltaics (OPV), as they offer improved efficiency and stability at relatively low costs [1,2]. As perovskite solar cells (PSCs) have become highly efficient in a very short time, much research effort is currently focused on the optimization of device interlayers [3,4,5].

Hybrid organic/inorganic perovskite solar cells (PSCs) are highly promising in photovoltaics because of their optoelectronic properties, including an excellent absorption coefficient, long carrier diffusion length, tuneable bandgap, and high mobility of the charge carriers [6,7,8]. Additionally, PSCs offer the advantage of a low-cost manufacturing process [9,10]. The power conversion efficiency (PCE) of PSCs with a regular (n-i-p type) structure has rapidly increased from its initial value of 3.8% [11] to the current 26.0% [12], which approaches the PCE of crystalline silicon solar cells [13,14]. However, the n-i-p type devices have some drawbacks, such as the high-temperature processing (500 °C) required for curing the mesoporous TiO_2_ of the electron transport material (ETM). In addition, the requirements for the hole transporting materials (HTMs), such as high hole conductivity and mobility, high thermal stability, and a proper energy level, reduce the number of potential candidates. To overcome these issues, an emerging inverted planar PSC (p-i-n type) structure was designed by Guo et al. [15]. This p-i-n type configuration shows many advantages compared to their regular counterparts, e.g., low-temperature processing and compatibility with large-scale fabrication [16,17], severely suppressed hysteresis effects, and good compatibility with the flexible substrate. Several studies have been recently published showing efficiencies near 25% [18,19].

Since HTMs and ETMs can aid the extraction of the free charge carriers at the corresponding electrodes [20], the exploration of new transporting materials has been one of the strategies to reduce the performance gap between both architectures, as they play a key role in the stability of the devices [21]. In inverted PSCs (iPSCs), the quality of the HTM film is crucial, since this affects the crystallinity and morphology of the perovskite film [22]. Conductive polymers such as polytriarylamine (PTAA) and poly(3,4-ethylenedioxythiophene)/poly(styrene sulfonic acid) (PEDOT:PSS) have been widely used as HTMs. They can modify the hydrophilic nature of the perovskite surface, reducing moisture [23], and enabling less sophisticated depositing methods [24] and lower-temperature annealing [25]. However, devices based on these polymeric HTMs still have some drawbacks that limit their future scaling up. The PEDOT:PSS-based devices yield moderate *V_oc_* values below 1.0 V, owing to the relatively low work function that does not match well with that of the perovskite [26]. These devices also present lower stability due to the hygroscopic nature and the acidity of PEDOT:PSS, accelerating the degradation of the perovskite (PVK) [27,28]. Finally, PEDOT:PSS can be partly dissolved by the precursor solvent of the PVK, resulting in nonhomogeneous films [23]. Doping of PEDOT:PSS to simultaneously improve its conductivity and reduce degradation issues has been explored by a few authors [29,30]. On the other hand, PTAA promotes higher *V_oc_* and power conversion efficiency (PCE) values. However, its high hydrophobic nature causes a more complex deposition method, which reduces the reproducibility of devices with higher charge recombination due to heterogeneous growth of the perovskite [31]. The high PTAA price, together with the need for doping [32], hinders its application in large-scale iPSC fabrication. To solve these concerns, several dopant-free small molecular HTMs have been synthesised using different types of central core moieties (e.g., thiophenyl, carbazole, truxene, pyrene, triphenylamine [20,31,33,34,35]). The utilization of small organic molecules as HTMs enables the production of extremely thin films (<20 nm), resulting in higher mobilities without the need for doping [36,37]. This can be accomplished using only a minimal quantity of the material. HTMs containing spiro-core structures extract holes more efficiently from the adjacent perovskite layer, as their perpendicular core geometry can prevent intermolecular π-π interactions, and form amorphous homogeneous layers [38,39,40,41]. Generally, the rigid 3D core structure is detrimental to hole mobility. However, the introduction of diverse functional groups could enhance the mobility of spiro-based thin films and modify the energy of the interface [42,43,44], which alters the crystal morphology of the perovskite layer.

Herein, in this work we present the design and synthesis of a novel spiro-type HTM (**Syl-SC**) by replacing one of the methoxyphenyl groups by ethoxytriisopropylsilane for each N atom, as shown in Figure 1. We have investigated the effects of the **Syl-SC** on the performance of p-i-n type PSCs based on CsFAMA triple-cation perovskite (Table 1). Reference cells were prepared using PEDOT:PSS and doped and undoped Spiro-OMeTAD (N2,N2,N2′,N2′,N7,N7,N7′,N7′-octakis(4-methoxyphenyl)-9,9′-spirobi[9H-fluorene]-2,2′,7,7′-tetramine) [45]. The incorporation of the triisopropylsilylether group improves the compound solubility [46], giving rise to uniform, hydrophobic, and high transmittance films, which can be easily deposited by spin-coating under environmental conditions.

After device optimization, we realized that the PCE of perovskite solar cells shows strong dependence on the thickness of the **Syl-SC** layer, which can be modulated by the precursor **Syl-SC** solution concentration. Impedance spectroscopy and transient optoelectronic measurements were carried out to further understand the interfacial charge recombination.

## 2. Materials and Methods

### 2.1. Materials

The PEDOT:PSS was purchased from Heraeus Deutschland GmbH and Co. KG, Hanau, Germany, Lead(II) bromide (PbBr_2_) and lead(II) iodide (PbI_2_) were acquired from TCI; formamidinium iodide (FAI) and methylammonium bromide (MABr) and were purchased from Dyenamo (99.99%). Cesium iodide (CsI) and all the anhydrous solvents, i.e., dimethyl sulfoxide (DMSO), dimethyl formamide (DMF), and chlorobenzene (CB), were purchased from Sigma Aldrich (St. Louis, MO, USA) and used without further purification. The patterned indium tin oxide (ITO, 15 Ω/square) glass substrates were provided by Xin Yan Technology Ltd., Hong Kong.

### 2.2. Deposition of the HTMs

The solution of PEDOT:PSS was spin-coated onto the ITO substrate at 5000 rpm for 45 s, and then annealed on a hot plate at 150 °C for 10 min in air. The **Syl-SC** was dissolved in chlorobenzene with three different concentrations (2, 0.8, and 0.5 mg/mL) and spin-coated on top of the ITO substrate at 3000 rpm for 30 s, and then annealed at 100 °C for 10 min in air.

### 2.3. Preparation of the Perovskite Triple-Cation Cs_0.05_FA_0.79_MA_0.16_Pb(I_0.85_Br_0.15_)_3_ (CsFAMA) Precursor Solution

The preparation of the perovskite precursor solution has been reported in our earlier works [45]: The perovskite precursors FAI (1.1 M), MABr (0.2 M), PbI_2_ (1.15 M), and PbBr_2_ (0.2 M) were dissolved in anhydrous DMF:DMSO (4:1 *v*/*v*). Then, 42 µL of a CsI stock solution (1.5 M in DMSO) was added to the mixed perovskite solution. The precursor solution was stirred and then filtered using a PTFE filter head (pore size of 0.22 µm).

### 2.4. Device Fabrication

The structure of the inverted planar PSCs (iPSCs) was ITO/HTM (PEDOT:PSS or **Syl-SC**)/CsFAMA/C_60_/BCP/Ag, as illustrated in Figure 1a. Prepatterned ITO glass substrates were sequentially cleaned with Mucasol solution (2% in deionized water), DI water, ethanol, and IPA in an ultrasonic bath for 10 min each. Firstly, 60 µL HTM (PEDOT:PSS or **Syl-SC**) solution was spin-coated onto the clean ITO substrate and annealed, then the ITOs were cooled down to room temperature and transferred into a N_2_-atmosphere glovebox. Next, the deposition of the triple-cation perovskite layer was achieved in a N_2_ glovebox by spin-coating onto the HTM film in a static antisolvent-assisted two-step procedure at 1000 rpm for 10 s, and 4000 rpm for 25 s, followed by the addition of 110 µL on the spinning substrate during the last 12 s for the antisolvent step. After that, the samples were annealed for 40 min at 100 °C. Afterwards, 20 nm of C_60_ and 8 nm of bathocuprine (BCP) were thermally evaporated successively on top of the perovskite layer as electron-selective layers. Lastly, a 100 nm Ag layer was deposited at low pressure (10^−6^ bar) to complete the device. The active area of all devices is 0.09 cm^2^.

### 2.5. Device Characterizations

The current density–voltage (*J–V*) curves were recorded using a Keithley source measure unit (Model 2400) as a voltage source, and a solar simulator ABET technologies, (model 11,000 class type A) as the light source. The measurements were registered under 1 Sun conditions (100 mW/cm^2^, AM 1.5 AG) calibrated with a silicon reference cell. The devices were sealed in a holder under a N_2_ atmosphere. The EQE spectra were recorded using a quantum efficiency measurement system from Lasing, S.A. (IPCE-DC, LS1109-232) with a Newport 2936-R power-meter unit. The light dependence of the open circuit voltage (*V_oc_*) and short-circuit current density (*J_sc_*) was established by measuring the *J–V* characteristics under different light intensities using a set of optical density filters.

The images of the surface morphology of the perovskite film were taken with a field emission scanning electron microscopy (FESEM, Thermo Fisher Scientific model Scios 2, Waltham, MA, USA). 

Photoinduced charge extraction (CE) and transient photovoltage (TPV) measurements were performed in open-circuit voltage equilibrium by illuminating the devices using a white light LED ring from LUXEON^®^, Lumileds, The Netherlands. The white LED ring is connected to a programmable power supply and a control box that controls the applied bias, providing different light intensities switched from open- to short-circuit states. All of the signals were recorded using a Yokogawa DLM2052 oscilloscope (Yokogawa Electric Corporation, Tokyo, Japan), which registers the voltage drops. In TPV measurements, the small light perturbation pulses were provided by a nanosecond PTI GL-3300 nitrogen laser with a 580 nm laser pulse wavelength (<100 ns pulses). Impedance spectroscopy (IS) measurements were carried out with a frequency range of 5 Hz–1 MHz at forward applied bias voltages of 0.75 V and an AC signal with 50 mV amplitude under 1 sun (AM 1.5 G) illumination using an HP-4193A impedance analyser, Hewlett-Packard Company, Palo Alto, CA, USA.

## 3. Results and Discussion

### 3.1. Synthesis and Photoelectrochemical Properties

Figure 1 depicts the synthesis pathway of the **Syl-SC** molecule. The spiro-type fluorine core [55] was incorporated through a Buchwald–Hartwing coupling [56,57] from the commercial 2,2′,7,7′-tetrabromo-9,9′-spirobi[fluorene], and the secondary amine **2**. The initial amine **1** was synthesized using a well-established procedure involving a nucleophilic substitution between 4-iodoanisole and ethanolamine [58]. The protection of the hydroxyl group with the triisopropylsilyl [56] produces the silyl amine **2**. **Syl-SC** was obtained with a good yield, and its chemical structure was characterized using nuclear magnetic resonance (NMR) spectroscopy (Appendix A). The thermal properties were investigated using thermogravimetric analysis (TGA) and differential scanning calorimetric (DSC) methods (Appendix A). From TGA analysis, it can be deduced that **Syl-SC** exhibited lower thermal stability compared to Spiro-OMeTAD, with a decomposed temperature at 123 °C, and 170 °C for the spiro (Appendix A), probably due to the N-ethoxytrialkylsislylether substituent incorporation. Considering that **Syl-SC** is a viscous oil under ambient conditions, and DSC measurements showed one glass transition at 171 °C for **Syl-SC** (Appendix A), we can assume a polymorphic behaviour, as well as for the Spiro-OMeTAD counterpart [59].

Differential pulse voltammetry (DPV) and cyclic voltammetry (CV) were used to analyse the electrochemical properties of **Syl-SC** (Appendix A) using tetrabutylammonium hexafluorophosphate (0.1 M) as the supporting electrolyte, a Pt counter electrode, a glassy carbon working electrode, and Ag/AgCl reference electrode. The redox potential and frontier orbitals are gathered in Appendix A. The highest occupied molecular orbital (HOMO) level was obtained from the half-wave potentials determined by CV, E_HOMO_ = −($E_{OX}^{I}$ + 4.8). Additionally, the optical band gap energy (E_g_) was utilized to estimate the lowest unoccupied molecular orbital (E_LUMO_ = E_g_ + E_HOMO_). Figure 1b illustrates the energy level alignment for the materials employed in the iPSCs. The low HOMO energy level of PEDOT:PSS usually yields a *V_oc_* between 0.85 and 1 V [30]. The **Syl-SC** has a deeper HOMO energy level than PEDOT:PSS, which better matches the valence band of CsFAMA. The high energy gap between the LUMO of **Syl-SC** to the CsFAMA conduction band indicates that **Syl-SC** can effectively block the injection of electrons from the perovskite to the anode, and suppress the current leakage. On the other hand, an effective electron transfer to the cathode is expected due to the proper alignment between the conduction band of perovskite and the lowest unoccupied molecular orbital (LUMO) levels of the electron transport layers (ETLs). The energy levels for ITO, PEDOT:PSS, CsFAMA, C_60_, BCP, and Ag were taken from the literature [21,31,60,61,62].

### 3.2. Electrical Characterization and Performance Analysis

The photovoltaic performance of iPSC devices containing **Syl-SC** as a dopant-free HTL was evaluated using the triple-cation perovskite CsFAMA as the absorber layer. Similar iPSCs using PEDOT:PSS as the HTM were fabricated as a reference, and compared with reference devices made of doped and undoped spiro. We studied the effects of **Syl-SC** thickness on the performance parameters of iPSCs by tuning the **Syl-SC** concentration in the chlorobenzene solution. The current density vs. voltage (*J*–*V*) curves of the best-performing iPSCs are shown in Figure 2a with PEDOT:PSS and three different **Syl-SC** concentrations, 0.5 mg/mL, 0.8 mg/mL, and 2 mg/mL, under 1 Sun illumination (AM 1.5G, 100 mW/cm^2^). The best-performing parameters of the optimization of **Syl-SC** concentration are summarized in Table 2. Average PCE and standard deviation values were calculated from over eight devices. Statistical results for the devices that contain PEDOT:PSS and **Syl-SC** with three different concentrations are shown in the Appendix A. We observed a decrease in the PCE, V_oc_, and fill factor (FF) of devices with a **Syl-SC** concentration of 2 mg/mL, whereas the *J_sc_* is quite similar to that of devices with 0.5 mg/mL and 0.8 mg/mL. PCE significantly improves by reducing the **Syl-SC** concentration from 2 mg/mL (PCE = 13.07%) to 0.5 mg/mL (PCE = 15.09%); however, we observe a lack of reproducibility in the devices fabricated with 0.5 mg/mL of **Syl-SC** solution. Notably, when using a **Syl-SC** concentration of 0.8 mg/mL, both *V_oc_* and *FF* were simultaneously improved, affording an increment of the PCE up to 15.77%.

The *J*–*V* curves of devices made with **Syl-SC** and PEDOT:PSS showed a low hysteresis effect; however, in devices with **Syl-SC,** the hysteresis reduced the FF and *V_oc_*, while in PEDOT-PSS-based devices, the hysteresis had a major effect on the FF. The hysteresis index (HI), HI = (PCE_reverse_ − PCE_forward_)/(PCE_reverse_) [63], was calculated to quantify the discrepancy between the two scanned efficiencies. The best **Syl-SC** device (0.8 mg/mL) shows the lowest hysteresis index in comparison with both PEDOT:PSS and spiro-based reference devices. The champion device based on **Syl-SC** exhibited the highest PCE of 15.77% (reverse scan), with a *J_sc_* of 20.00 mA/cm^2^, a *V_oc_* of 1.006 V, and an FF of 80.10%. On the other hand, the champion device based on PEDOT:PSS exhibited a PCE of 14.76% (reverse scan), with a *J_sc_* of 21.37 mA/cm^2^, a *V_oc_* of 0.866V, and an FF of 79.70%, values that are higher than the references made with doped and undoped spiro in our previous work [45]. Thus, we have continued our work, taking as a sole reference the device made with PEDOT:PSS. The higher PCE of the iPSC with **Syl-SC** (0.8 mg/mL) can be mainly attributed to its higher *V_oc_* compared to the device with PEDOT:PSS (1.006 and 0.866 V, respectively). However, the iPSC with PEDOT:PSS exhibits a higher *J_sc_* than that of the **Syl-SC**-based device (21.37 and 19.57 mA/cm^2^, respectively). To validate the *J_sc_* calculated from *J*–*V* curves, external quantum efficiency (EQE) measurements were performed on devices with **Syl-SC** and PEDOT:PSS. Figure 2c displays the EQE spectra and the integrated *J_sc_* of the PSCs. The integrated *J_sc_*, calculated from the EQE, is 21.45 mA/cm^2^ for PEDOT:PSS and 19.25 mA/cm^2^ for **Syl-SC**, which fits with the current density extracted from the *J*–*V* curve. 

### 3.3. Morphological Characterization of the Films

Since the hole-transporting layer can influence the crystallinity and morphology of perovskite film, surface modifications of the HTL layer have been investigated. Images of the water droplets on the surface of PEDOT:PSS and **Syl-SC** are shown in Appendix A. The contact angle (CA) test demonstrated the hydrophilic nature of the PEDOT:PSS film, with a CA of 16°, whereas the **Syl-SC** film exhibited a higher contact angle of 73°. The higher wettability of the bottom layer promotes better spreading of the perovskite precursor solution during the spin-coating process. Nevertheless, some studies have reported that a relatively hydrophobic surface promotes the formation of high-quality polycrystalline films compared to those deposited on a hydrophilic surface [22]. In addition, the effects of HTMs on the surface morphology of CsFAMA films were characterized by a field emission scanning electron microscope (FESEM). The FESEM images show a highly smooth surface for the PEDOT:PSS film (Appendix A), while **Syl-SC** forms a structured domain surface (Appendix A). The FESEM images of CsFAMA grown on PEDOT:PSS and **Syl-SC** (0.8 mg/mL) are shown in Figure 3a,b, respectively. It is noteworthy that perovskite films presented a smooth surface and full coverage in both HTMs, and pinholes between grain boundaries were not observed. As evidenced by the FESEM surface images, the CsFAMA film grown on **Syl-SC** showed a smaller grain size than that of CsFAMA deposited on PEDOT:PSS. The grain size distribution analysis (Figure 3c) reveals a perovskite average grain size difference of more than 100 nm between the two studied bottom substrates. The **Syl-SC** film promotes the formation of smaller CsFAMA crystals (approximately 163.9 nm in diameter) compared to PEDOT:PSS (approximately 271.6 nm).

The larger crystals observed on PEDOT:PSS indicated a better growth of the perovskite film, which can explain the higher *J_sc_* value for the PEDOT-PSS-based device [22,31]. Figure 3d displays a cross-sectional view obtained by FESEM of the iPSC based on **Syl-SC**. The image shows a compact and clear multilayered structure with well-defined interfaces. The cross-sectional FESEM images show that **Syl-SC** promotes a proper crystallization of CsFAMA. The thickness of all layers that integrate the device was measured from the FESEM images (see Appendix A).

### 3.4. Charge Recombination Characterization

To assess the influence of the **Syl-SC** interlayer on the interfacial charge recombination in devices, we analysed the light intensity dependence of *V_oc_* and *J_sc_*. Figure 4a shows the variation of *J_sc_* against light intensity (*P_light_*) fitted by the power-law function $J_{SC}={P_{light}}^{\alpha }$, where α represents the second-order bimolecular recombination degree. A power (α) value ~1 suggests that the charge recombination mechanism can be mostly assigned to the monomolecular recombination processes, whereas the bimolecular recombination is negligible under short-circuit conditions [64,65]. The estimated α value of samples with **Syl-SC** and PEDOT:PSS were 0.95 and 0.90, respectively, which suggested a low contribution of bimolecular recombination. However, the slightly higher α value indicates that **Syl-SC** can better reduce the bimolecular recombination than PEDOT:PSS. Figure 4b shows the light intensity dependence of *V_oc_*. The semilogarithmic *V_oc_* vs. *P_light_* plot was fitted by the equation $V_{OC}=n_{id}\;(\;kT/q\;)\;Ln\;{(P}_{light})+C$, where $n_{id}$ is the ideality factor, k is the Boltzmann constant, T is the temperature, q is the elementary charge and C is a fitting parameter. The expected n value ranges between 1 and 2 ($1\leq n_{id}\leq 2$); the bimolecular recombination is dominating if the $n_{id}$ value approaches 1, while the monomolecular recombination mechanisms are responsible for the charge recombination (e.g., trap-assisted and geminate recombination) when $n_{id}$ values are close to 2 [66,67]. The devices with PEDOT:PSS and **Syl-SC** exhibited similar $n_{id}$ close to 1 (1.05 and 1.10), which indicates both HTMs can suppress the monomolecular recombination under open-circuit conditions. These results suggest that **Syl-SC** enables perovskite films of good quality with low defects or impurities at the HTM/CsFAMA interface and at grain boundaries, which reduces the monomolecular recombination [68].

We also carried out electrochemical impedance spectroscopy (EIS) measurements to investigate charge recombination. Figure 4c displays the Nyquist plots of devices with PEDOT:PSS and **Syl-SC** measured under 1 Sun illumination at 0.75 V bias. Both iPSCs presented one typical semicircle shape corresponding to the resistor/capacitor circuit [69]. The EIS responses were interpreted using an equivalent-circuit model with one external series resistance (R_series_) and one resistor/capacitor (RC) element, as shown in the inset of Figure 4c. The fitting parameters of the circuit elements are summarized in Appendix A. Since the iPSCs have similar structures, and the only difference lies in the HTM layer, the changes in the EIS measurements are attributed to the HTM/CsFAMA interface. The R_series_ is associated with the contact resistances and sheet resistance of ITO, and can be estimated from the high-frequency range. The similar R_series_ of iPSCs with PEDOT:PSS and **Syl-SC** (7.18 and 7.30 Ω, respectively) indicate that there are no major differences in the hole-extraction dynamics between devices with the two HTMs. On the other hand, the characteristic semicircle describes the electrochemical behaviour of devices, which involves geometrical capacitance (C1) and interfacial charge recombination resistances (R1). The recombination resistance (R1 in Appendix A) is 35.01 Ω and 25.11 Ω for devices with **Syl-SC** and PEDOT:PSS, respectively. The larger recombination resistance implies lower recombination losses; thus, **Syl-SC** can better suppress the interfacial charge recombination.

To obtain further insight into the effects of **Syl-SC** on the charge recombination dynamics, we carried out charge extraction (CE) and transient photovoltage (TPV) measurements under standard device PV operating conditions in terms of light intensity and applied voltage [70], which is important for PSCs that display light intensity-dependent properties [71,72,73]. As a large modulation transient method, CE is a measurement that allows extraction of all the charges present in the solar cell once the *V_oc_* of the device is stabilized, and while the illumination is switched off. After a short circuit of the solar cell, the resultant transient discharging current is measured through a small external load resistor. Figure 5a shows the charge extraction of the device obtained applying different open circuit voltages under illumination. The charge density exhibited two different regimes, linear and exponential. The linear trend in the voltage region 0–0.8 V is attributed to accumulated charge at the electrodes (geometrical capacitance). On the other hand, after 0.8 V, the charge carrier density exhibited an exponential dependency on the applied voltage, and is related to the accumulated charge at the HTM/perovskite/ETM interfaces (chemical capacitance). The charge density within the bulk of the device is estimated by subtracting the geometrical capacitance from the CE data. The charge density at the bulk (solid lines in Figure 5a) shows a more pronounced slope at 0.7–0.85 V for the PEDOT:PSS, and at 1 V for **Syl-SC**. The better energy alignment between the HOMO of the **Syl-SC** and the perovskite valence band (VB) would be the reason for these differences in the charge vs. voltage. As previously reported [74], the resulting *V_oc_* arises from the respective HOMO energy level [75], as well as the disorder of the density of states and the recombination constant [72,76]. So thus, we analyse the interfacial carrier losses using TPV measurements in the different solar cell devices. Appendix A shows the variation of the carrier lifetime ($\tau_{∆n}$) as a function of the *V_oc_*. To analyse the recombination kinetics more effectively, we compare the carrier lifetime (extracted from the TPV plot) as a function of charge density (extracted from the CE plot), as shown in Figure 5b. Thus, we compare the differences in carrier lifetime under a determinate charge value (vertical solid line at 4 × 10^−8^ C/cm^2^), as shown in the inset of Figure 5b. The recombination in the devices with PEDOT:PSS is one order of magnitude faster than that of devices with **Syl-SC**, which explains the differences in *V_oc_* values. Thus, **Syl-SC** can better reduce the charge recombination than that PEDOT:PSS. This result agrees with those obtained from light intensity-dependence of *V_oc_* and *J_sc_*, and impedance spectroscopy analyses.

## 4. Conclusions

To summarize, we have successfully synthesized a new hole-transporting small molecule, **Syl-SC**, and used it as a dopant-free hole-transport layer (HTL) for inverted planar perovskite solar cells (PSCs) under ambient conditions. We showed that optimizing the thickness of the HTL is an effective approach to enhance the device performance in the inverted configuration. The champion device with a PCE of 15.77% is made of a 15 nm ultrathin HTL obtained from a **Syl-SC** precursor solution of 0.8 mg/mL. Compared to the control PEDOT:PSS HTL and references with undoped and doped Spiro-OMeTAD, the **Syl-SC** HTL improved the *V_oc_* of iPSCs devices because of several reasons. This improvement can be attributed to the deeper HOMO energy level of **Syl-SC**, which better aligns with the valence band of the CsFAMA. Further analysis using impedance spectroscopy and transient optoelectronic measurements revealed that the **Syl-SC** HTL significantly reduced interfacial charge collection, improved fill factor (FF), and overall device efficiency. These findings demonstrate that modifying the well-known Spiro-OMeTAD through side-chain engineering is a promising strategy for developing efficient hole-transporting materials (HTMs), without the need for chemical dopants.

## Data Availability

Not applicable.

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
