# Peer review of "Novel Spiro-Core Dopant-Free Hole Transporting Material for Planar Inverted Perovskite Solar Cells"

_nanomaterials, 2023, doi:10.3390/nano13142042_

Round 1

Reviewer 1 Report

The authors designed and synthesized of a novel spiro-type HTM (Syl-SC) by replacing one of the methoxyphenyl groups by ethoxytriisopropylsilane for each N atom. Syl-SC was employed as hole transport layer for the CsFAMA triple cation perovskite solar cells, which delivered enhanced photovoltaic when compared to the PEDOT:PSS hole transport layer. In my point of view, this manuscript can be accepted for publication after minor revision as indicated in the following:

1. In the introduction part, some literatures representing the latest rapid development in the field of perovskite photovoltaic technology should be cited

2. How about the device stability?

Author Response

  1. In the introduction part, some literatures representing the latest rapid development in the field of perovskite photovoltaic technology should be cited

We acknowledge the positive feedback from the referee and the corresponding suggestions. Therefore, and in order to address this comment, we have added 4 new references in the introduction related to recent articles dealing with the modification of the interfaces in PSC and the latest efficiency records. The references are listed below, and have been cited in the following paragraph:

On page 1, line 36:

As perovskite solar cells (PSCs) have become highly efficient in a very short time, many research effort is focus nowadays on the optimization of device interlayers.[3–5]

On page 1, line 44:

The power conversion efficiency (PCE) of PSCs with a regular (n-i-p type) structure has rapidly increased from its initial value of 3.8%[11] to the current 26.0%[12] which approaches the PCE of crystalline silicon solar cells.[13,14]

References

  1. Wu, Y.; Zhu, H.; Yu, B. Bin; Akin, S.; Liu, Y.; Shen, Z.; Pan, L.; Cai, H. Interface Modification to Achieve High-Efficiency and Stable Perovskite Solar Cells. Chemical Engineering Journal 2022, 433, doi:10.1016/j.cej.2022.134613.
  2. Akman, E.; Shalan, A.E.; Sadegh, F.; Akin, S. Moisture-Resistant FAPbI3 Perovskite Solar Cell with 22.25 % Power Conversion Efficiency through Pentafluorobenzyl Phosphonic Acid Passivation. ChemSusChem 2021, 14, 1176–1183, doi:10.1002/cssc.202002707.
  3. Hao, M.; Duan, T.; Ma, Z.; Ju, M.G.; Bennett, J.A.; Liu, T.; Guo, P.; Zhou, Y. Flattening Grain-Boundary Grooves for Perovskite Solar Cells with High Optomechanical Reliability. Advanced Materials 2023, doi:10.1002/adma.202211155.
  4. Cell-Pv-Eff-Emergingpv Available online: https://www.nrel.gov/pv/cell-efficiency.html (accessed on 29 June 2023).

  1. How about the device stability?

We agree with the referee that stability is an important factor in the development of perovskite solar cell. However, in this work, we have focused on the synthesis of novel molecules and their application in devices to check their efficiency and establish a relationship with the interfacial charge transfer. This has been done with the final aim of the analysis understanding how the ethoxytriisopropylsilane substitution affects the interfacial charge extraction and transfer.

As we are aware of the importance of studying the stability of devices, we are conducting it for an independent work through different temperature, humidity and illumination conditions to determine the decomposition mechanism of perovskite crystal grown onto Syl-SC.

Reviewer 2 Report

The manuscript provides the synthesis and performance evaluation of a novel hole-transporting material (HTM) called Syl-SC for inverted planar perovskite solar cells (iPSCs). The researchers compared the performance of iPSCs using Syl-SC with iPSCs using PEDOT:PSS, a commonly used HTM in perovskite solar cells. This manuscript is interesting and should be published after a few major concerns as follow:

1.      The current research objectives of this manuscript and the gap in the existing literature could be more clearly articulated on PSCs.

2.      The introduction makes several claims and statements without providing specific citations, especially about Silicon solar cell in line 34. Please cite these article (DOI:10.1021/acsaem.2c01194 and https://doi.org/10.1002/er.7924) here.

3.      Also, it would be beneficial to provide a more direct comparison between these different HTMs, highlighting their respective advantages and disadvantages. This would provide a clearer context for the need for the novel Syl-SC HTM.

4.      In the introduction, a few sentences are overly long and complex, which could make it challenging for readers to understand the main points effectively. So, please look at this issue.

5.      The picture of Scheme 1 could be better.

6.       The spin coating of perovskite materials was accomplished in the static or dynamic situation of spin-coater?

7.      Why BCP layer is used? If it is used for an insulating layer then 8 nm seems thick. Explain

8.      By the way what is the novelty of this work?

9.      For broad readership cite these articles in the introduction for photodetection and Organic Solar Cells.

A.      doi: https://doi.org/10.1002/adom.202201396

B.      doi: https://doi.org/10.1002/solr.202300143

10.  All figures need serious intentions to improve.

11.  Stability of PSC is a major issue nowadays. Authors should provide some experimental evidence for the stability of devices regarding PCE. 

Author Response

  1. The current research objectives of this manuscript and the gap in the existing literature could be more clearly articulated on PSCs.

We appreciate the positive comment of the reviewer and feedback that will improve the quality of the manuscript. Therefore, and in order to address this comment, we have added 4 new references in the introduction related to recent articles dealing with the modification of the interfaces in PSC and the latest efficiency records. The references are listed below, and have been cited in the following paragraph:

On page 1, line 36: As perovskite solar cells (PSCs) have become highly efficient in a very short time, many research effort is focus nowadays on the optimization of device interlayers.[3–5]

On page 1, line 44: The power conversion efficiency (PCE) of PSCs with a regular (n-i-p type) structure has rapidly increased from its initial value of 3.8%[11] to the current 26.0%[12] which approaches the PCE of crystalline silicon solar cells.[13,14]

References

  1. Wu, Y.; Zhu, H.; Yu, B. Bin; Akin, S.; Liu, Y.; Shen, Z.; Pan, L.; Cai, H. Interface Modification to Achieve High-Efficiency and Stable Perovskite Solar Cells. Chemical Engineering Journal 2022, 433, doi:10.1016/j.cej.2022.134613.
  2. Akman, E.; Shalan, A.E.; Sadegh, F.; Akin, S. Moisture-Resistant FAPbI3 Perovskite Solar Cell with 22.25 % Power Conversion Efficiency through Pentafluorobenzyl Phosphonic Acid Passivation. ChemSusChem 2021, 14, 1176–1183, doi:10.1002/cssc.202002707.
  3. Hao, M.; Duan, T.; Ma, Z.; Ju, M.G.; Bennett, J.A.; Liu, T.; Guo, P.; Zhou, Y. Flattening Grain-Boundary Grooves for Perovskite Solar Cells with High Optomechanical Reliability. Advanced Materials 2023, doi:10.1002/adma.202211155.
  4. Cell-Pv-Eff-Emergingpv Available online: https://www.nrel.gov/pv/cell-efficiency.html (accessed on 29 June 2023).

Moreover, we have inserted in the introduction a comparative table that summarize the advantages and disadvantages of the Syl-SC compound respect to the usual HTM references (PEDOT:PSS, PTAA and Spiro-OMeTAD).

On page 3: Table 1. Comparison of the advantages and disadvantages of PEDOT:PSS, PTAA, SpirO-MeTAD and Syl-SC.

  1. The introduction makes several claims and statements without providing specific citations, especially about Silicon solar cell in line 34. Please cite these article (DOI:10.1021/acsaem.2c01194 and https://doi.org/10.1002/er.7924) here.

We acknowledge the suggestion and have cited in the manuscript the suggested publication:

On page 1, line 44: The power conversion efficiency (PCE) of PSCs with a regular (n-i-p type) structure has rapidly increased from its initial value of 3.8%[11] to the current 26.0%[12] which approaches the PCE of crystalline silicon solar cells.[13,14].

References

  1. Rehman, M.A.; Park, S.; Khan, M.F.; Bhopal, M.F.; Nazir, G.; Kim, M.; Farooq, A.; Ha, J.; Rehman, S.; Jun, S.C.; et al. Development of Directly Grown-Graphene–Silicon Schottky Barrier Solar Cell Using Co-Doping Technique. Int J Energy Res 2022, 46, 11510–11522, doi:https://doi.org/10.1002/er.7924.
  2. Kadam, K.D.; Rehman, M.A.; Kim, H.; Rehman, S.; Khan, M.A.; Patil, H.; Aziz, J.; Park, S.; Abdul Basit, M.; Khan, K.; et al. Enhanced and Passivated Co-Doping Effect of Organic Molecule and Bromine on Graphene/HfO2/Silicon Metal–Insulator–Semiconductor (MIS) Schottky Junction Solar Cells. ACS Appl Energy Mater 2022, 5, 10509–10517, doi:10.1021/acsaem.2c01194.

  1. Also, it would be beneficial to provide a more direct comparison between these different HTMs, highlighting their respective advantages and disadvantages. This would provide a clearer context for the need for the novel Syl-SC HTM.

We appreciate this suggestion and thus have added a comparative table in the introduction to summarize the advantages and disadvantages of the use of Syl-SC in comparison to the usual HTM references (PEDOT:PSS, PTAA and Spiro-OMeTAD).

On page 3:

Table 1. Comparison of the advantages and disadvantages of PEDOT:PSS, PTAA, SpirO-MeTAD and Syl-SC.

Table 1. Comparison of the advantages and disadvantages of PEDOT:PSS, PTAA, SpirO-MeTAD and Syl-SC

Ref

[47]

[23]

[48]

[26]

[49]

[32]

[48]

[50]

[50]

[51]

[This work]

Dis advantages

§  The high hygroscopic nature can favor the capture of humidity, leading to device degradation. Its acidity can also corrode ITO

§  It can be partly dissolved by perovskite precursor solvent

§  PEDOT:PSS films may suffer from poor optical transparency, reducing the amount of light reaching the active perovskite layer

§  Relatively low work function

§  The high hydrophobic nature causes a more complex deposition method, increasing the manufacturing challenges It can be partly dissolved by perovskite precursor solvent.

§  To increase the hole mobility, the addiction of dopants into the HTL is necessary

§  High cost and inappropriate for the fabrication of large-area devices

§  Spiro are unstable at high temperatures (60–120 â—¦C) for long times It can be partly dissolved by perovskite precursor solvent

§  The addition of dopants is necessary due to the low conductivity and mobility

§  To successful PV results is necessary a thick layer around 200-300 nm without doping

§  Spiro are unstable at high temperatures (60–120 â—¦C) for long times It can be partly dissolved by perovskite precursor solvent

§  The addition of dopants is necessary due to the low conductivity and mobility

§  To successful PV results is necessary a thick layer around 200-300 nm without doping

Ref

[52]

[53]

[49]

[54]

[51]

[47]

[55]

[This work]

Advantages

§  Good hole extraction and transport properties

§  Solution-processed allowing low-cost and large-scale production

§  Well-established material in the perovskite solar cells field, providing a benchmark for comparison

§  PTTA promotes higher PCE values

§  Good thermal stability

§  Form a favorable interface with perovskite, promoting efficient charge transfer and reducing recombination losses

§  Provides a good energy level alignment at the perovskite/HTM interface facilitating the charge extraction

§  Can prevent intermolecular p-p interactions

§  High solubility, film formability, proper ionization potential, matched absorption spectrum, and smooth solid-state morphology

§  Easy to synthesize and solution processed

§  Appropriate energy levels for transferring the holes and blocking electrons

§  Good PCE results are obtained using a minimal amount of compound without dopants

§  Form ultrathin (<20 nm), homogenous, and high transmittace films under environmental conditions

§   

Material

PEDOT:PSS

PTTA

Spiro-OMeTAD

Syl-SC

References:

  1. Vaghi, L.; Rizzo, F. The Future of Spirobifluorene-Based Molecules as Hole-Transporting Materials for Solar Cells. Solar RRL 2023, doi:10.1002/solr.202201108.
  2. Loizos, M.; Tountas, M.; Tzoganakis, N.; Chochos, C.L.; Nega, A.; Schiza, A.; Polyzoidis, C.; Gregoriou, V.G.; Kymakis, E. Enhancing the Lifetime of Inverted Perovskite Solar Cells Using a New Hydrophobic Hole Transport Material. Energy Advances 2022, 1, 312–320, doi:10.1039/d2ya00067a.
  3. Wang, Y.; Duan, L.; Zhang, M.; Hameiri, Z.; Liu, X.; Bai, Y.; Hao, X. PTAA as Efficient Hole Transport Materials in Perovskite Solar Cells: A Review. Solar RRL 2022, 6.
  4. Jena, A.K.; Numata, Y.; Ikegami, M.; Miyasaka, T. Role of Spiro-OMeTAD in Performance Deterioration of Perovskite Solar Cells at High Temperature and Reuse of the Perovskite Films to Avoid Pb-Waste. J Mater Chem A Mater 2018, 6, 2219–2230, doi:10.1039/c7ta07674f.
  5. Wang, C.; Hu, J.; Li, C.; Qiu, S.; Liu, X.; Zeng, L.; Liu, C.; Mai, Y.; Guo, F. Spiro-Linked Molecular Hole-Transport Materials for Highly Efficient Inverted Perovskite Solar Cells. Solar RRL 2020, 4, doi:10.1002/solr.201900389.
  6. Hu, L.; Li, M.; Yang, K.; Xiong, Z.; Yang, B.; Wang, M.; Tang, X.; Zang, Z.; Liu, X.; Li, B.; et al. PEDOT:PSS Monolayers to Enhance the Hole Extraction and Stability of Perovskite Solar Cells. J Mater Chem A Mater 2018, 6, 16583–16589, doi:10.1039/c8ta05234d.
  7. Urieta-Mora, J.; García-Benito, I.; Molina-Ontoria, A.; Martín, N. Hole Transporting Materials for Perovskite Solar Cells: A Chemical Approach. Chem Soc Rev 2018, 47, 8541–8571.
  8. Ke, Q. Bin; Wu, J.R.; Lin, C.C.; Chang, S.H. Understanding the PEDOT:PSS, PTAA and P3CT-X Hole-Transport-Layer-Based Inverted Perovskite Solar Cells. Polymers (Basel) 2022, 14.
  9. Hatamvand, M.; Vivo, P.; Liu, M.; Tayyab, M.; Dastan, D.; Cai, X.; Chen, M.; Zhan, Y.; Chen, Y.; Huang, W. The Role of Different Dopants of Spiro-OMeTAD Hole Transport Material on the Stability of Perovskite Solar Cells: A Mini Review. Vacuum 2023, 214.

  1. In the introduction, a few sentences are overly long and complex, which could make it challenging for readers to understand the main points effectively. So, please look at this issue.

We thank the reviewer for this comment and have rewritten several sentences to make the introduction clearer and more readable.

Page 1, paragraph 2:Hybrid organic-inorganic perovskite solar cells (PSCs) are highly promising in photovoltaics due to their optoelectronic properties including excellent absorption coefficient, long carrier diffusion length, high charge carrier mobility and widely tunable bandgap;[1–3]. Additionally, PSCs offer the advantage of a low-cost manufacturing process [9,10].

Page 2, paragraph 3: Conductive polymers such as polytriarylamine (PTAA) and poly(3,4-ethylenedioxythiophene)/poly(styrene sulfonic acid) (PEDOT:PSS) have been widely used as HTMs. They can modify the hydrophilic nature of the

perovskite surface, reducing moisture[23] and  enabling less sophisticated depositing methods [24] and lower-temperature annealing[25].

Page 2, paragraph 3: On the other hand, PTAA promotes higher Voc and power conversion efficiency (PCE) values However, its high hydrophobic nature causes a more complex deposition method, which reduces the reproducibility of devices with higher charge recombination due to heterogeneous growth of the perovskite [31].

Page 2, paragraph 3: Generally, the rigid 3D core structure is detrimental to hole mobility. However, the introduction of diverse functional groups could enhance

the mobility of spiro-based thin films and modify the energy of the interface[42–44], which alters the crystal morphology of the perovskite layer.

  1. The picture of Scheme 1 could be better.

Thank you for your comment. The Scheme 1 has been modified as follows:

  1. The spin coating of perovskite materials was accomplished in the static or dynamic situation of spin-coater?

The spin coating was done in static situation assisted with dynamic antisolvent process to grow perovskite. We have indicated it in the Experimental Section:

Page 4, line 131: Next, the triple-cation perovskite layer was deposited in an N2 glovebox by spin coating onto the HTM film in a static two-step procedure at 1000 rpm for 10 s and 4000 rpm for 25 s. A 110 µl volume of chlorobenzene was dropped on the spinning substrate during the last 12 s, and then, the samples were annealed at 100 °C for 40 min.

  1. Why BCP layer is used? If it is used for an insulating layer then 8 nm seems thick. Explain.

Thank you so much for the comment. As shown in Figure 1b, it can be seen that the energy level, specifically, the highest occupied molecular orbital (HOMO) of C60, does not match with the perovskite valence band. This has implications in the hole blocking ability of C60 which results in the recombination between the electron and hole and decrease of the generated photocurrent. The addition of the BCP layer, with a deeper HOMO level, has the objective of blocking the holes in the perovskite to avoid the recombination and leakage current. The thickness of BCP is optimized according to its conductivity and coverage. The value of 8 nm is the optimized thickness in our lab, and consistent with the published papers (DOI: 10.1039/d0ee03807e; DOI: 10.1039/c9ee02268f).

  1. By the way what is the novelty of this work?

In this work we report the replacement of the methoxyphenyl groups of the widely used Spiro-OMeTAD molecule ethoxytriisopropylsilane for each N atom. This is the first report to use ethoxytriisopropylsilane in combination with the spiro core. The results showed that the new molecule has good solubility that allows to grow uniform films, as well as suitable energy level alignment with perovskite to improve the open-circuit voltage (Voc) of the device.

We report the organic synthesis and device preparation along the photophysical characterization to unravel the charge dynamics taking place in the devices. And this has been done after the increase in the efficiency of the devices because of the application of the Syl-C in comparison to the popular Spiro-OMeTAD.

  1. For broad readership cite these articles in the introduction for photodetection and Organic Solar Cells.

We appreciate these comments and for enhancing the quality of the introduction, we have cited the suggested references adding another one about ternary solar cell:

On page 1, line 34: Ternary organic solar cells (TOSCs) are being studied in the field of organic photovoltaics (OPV) as they offer improved efficiency and stability at relatively low costs.[1,2]

References:

  1. Sánchez, J.G.; Cabrera-Espinoza, A.; Martínez-Ferrero, E.; Delgado, J.L.; Palomares, E. Chalcogen-Substituted PCBM Derivatives as Ternary Components in PM6:Y6 Solar Cells. Mater Adv 2022, 3, 1071–1078, doi:10.1039/D1MA00925G.
  2. Huang, X.-M.; Chen, N.; Ye, D.-N.; Zhong, A.-G.; Liu, H.; Li, Z.; Liu, S.-Y. Structurally Complementary Star-Shaped Unfused Ring Electron Acceptors with Simultaneously Enhanced Device Parameters for Ternary Organic Solar Cells. Solar RRL 2023, 7, 2300143, doi:https://doi.org/10.1002/solr.202300143.

Regarding the suggestion to cite the reference https://doi.org/10.1002/adom.202201396, we are puzzled about this because the topic “High-Performance Visible to Near-Infrared Broadband Bi2O2Se Nanoribbon Photodetectors” does not fit with the research proposed in our manuscript. Therefore, we have not included it.

  1. All figures need serious intentions to improve.

Thank you for your comment. All the figures have been changed to improve their quality image.

  1. Stability of PSC is a major issue nowadays. Authors should provide some experimental evidence for the stability of devices regarding PCE.

We agree with the referee that stability is an important factor in the development of perovskite solar cell. However, in this work, we have focused on the synthesis of novel molecules and their application in devices to check their efficiency and establish a relationship with the interfacial charge transfer. This has been done

with the final aim of the analysis understanding how the ethoxytriisopropylsilane substitution affects the interfacial charge extraction and transfer.

As we are aware of the importance of studying the stability of devices, we are conducting it for an independent work through different temperature, humidity, and illumination conditions to determine the decomposition mechanism of perovskite crystal grown onto Syl-SC.

Round 2

Reviewer 2 Report

The authors made a great effort in responding to all the questions raised in the first revision. Therefore, I recommend this manuscript for final publication.